# Comparison of Two Commercial ELISA Kits for the Detection of Anti-Dengue IgM for Routine Dengue Diagnosis in Laos

**DOI:** 10.3390/tropicalmed4030111

**Published:** 2019-07-25

**Authors:** Yixiao Lu, Onanong Sengvilaipaseuth, Anisone Chanthongthip, Ooyanong Phonemixay, Manivanh Vongsouvath, Phonelavanh Phouminh, Stuart D. Blacksell, Paul N. Newton, Audrey Dubot-Pérès

**Affiliations:** 1Lao-Oxford-Mahosot Hospital-Wellcome Trust Research Unit (LOMWRU), Microbiology Laboratory, Mahosot Hospital, Vientiane 01000, Laos; 2Mahidol-Oxford Tropical Medicine Research Unit, Faculty of Tropical Medicine, Mahidol University, Bangkok 10400, Thailand; 3Centre for Tropical Medicine and Global Health, Nuffield Department of Clinical Medicine, University of Oxford, Churchill Hospital, Oxford OX3 7LJ, UK; 4Unité des Virus Émergents (UVE: Aix-Marseille Univ–IRD 190–Inserm 1207–IHU Méditerranée Infection), 13005 Marseille, France

**Keywords:** dengue, MAC-ELISA, serology, Laos, IgM

## Abstract

The endemicity of Dengue virus (DENV) infection remains a major public health problem in Lao PDR. In this study, we compared two commercial anti-dengue IgM ELISA kits, Panbio^®^ Dengue IgM Capture ELISA (Panbio Kit, Alere, Waltham, MA, USA) and DEN Detect^TM^ MAC-ELISA (InBios kit, InBios International, Inc., Seattle, WA, USA), in the context of diagnosis of patients admitted to hospital with clinical dengue presentation. Two panels of paired blood samples were tested. Panel A was composed of 54 dengue confirmed patients (by DENV real-time RT-PCR) and 11 non-dengue dengue patients (other infections confirmed by corresponding PCR results). Panel B included 74 patients randomly selected from consecutive patients admitted to Mahosot Hospital in 2008 with suspicion of dengue fever according to WHO criteria. Results from panel A showed significantly better sensitivity for Panbio kit (64.8%; 95%CI: 50.6–77.3%) than for InBios kit (18.5%; 95%CI: 9.3–31.4%) when testing admission sera. Sensitivity was increased for both kits when combining results from admission and convalescent sera. Concordant results were obtained from panel B with fair agreement (κ = 0.29) between both kits when testing single admission samples, and moderate agreement (κ = 0.5) when combining results from admission and convalescent sera.

## 1. Introduction

Dengue virus (DENV) infection is a mosquito-borne disease, mainly transmitted by *Aedes aegypti*. Dengue viruses are enveloped single-stranded RNA viruses distributed into four antigenically related serotypes (DENV1, DENV2, DENV3, and DENV4). DENV has reemerged in recent decades in over 100 tropical and subtropical countries with an estimation of 390 million infections occurring per year [1,2]. The most common form of the disease is dengue fever (DF) with mild manifestation, but in rare cases, it can develop into severe dengue hemorrhagic fever (DHF) or dengue shock syndrome (DSS) and may be fatal (an estimated 500,000 severe dengue cases occur yearly with 2.5% mortality) [1]. Dengue fever shares similar clinical presentation (fever, headache, retro-orbital, myalgia, nausea, rash) with other infections (leptospirosis, rickettsiosis, malaria, other arboviruses) occurring in the same areas [3,4]. Therefore, they are difficult to clinically distinguish from each other.

In order to increase the sensitivity of diagnosis, it is recommended to include both antigen/viral nucleic acid detection and antibodies detection in the protocol for the laboratory diagnosis of dengue infections [5]. Anti-dengue IgM antibodies are detectable from 5 days after the fever onset, for up to 2-3 months in a primary infection. MAC-ELISA is the recommended serological technique with good sensitivity and specificity, ease of use, and good performance on a single acute-phase blood sample [6,7]. Nowadays, several MAC-ELISA kits are commercially available for dengue infection diagnosis, but very few of them were evaluated in the context of routine use and the sensitivities and specificities of kits are difficult to compare due to differences in the choice of the reference test, samples used, and the incidence of dengue infection in the tested population. Two commercial dengue MAC-ELISA kits, the Panbio^®^ Dengue IgM Capture ELISA (Alere now Abbott, MA, USA) and the InBios DEN Detect^TM^ MAC-ELISA (InBios International, Inc., Seattle, WA, USA) from major commercial manufacturers were selected for this study. The Panbio kit was comparatively well evaluated in previous studies (the sensitivities and specificities were, 83.2% and 87.8% in Blacksell et al. [8], 87% and 96% in Groen et al. [9], 96.8% and 99.4% in Vazquez et al. [10], 87.6% and 88.1% in Pal et al. [11]), but there are only two publications concerning the evaluation of the InBios kit with limited sample size and unclear origins of the sera [12,13]. In this study, we compared the effectiveness of the two kits for the detection of anti-dengue IgM in the context of hospital diagnosis in Lao PDR. The comparison proceeded with two approaches; (1) using patients’ samples selected on the basis of positive molecular test results; (2) using randomly selected samples from patients admitted to Mahosot Hospital for suspicion of dengue infection.

## 2. Materials and Methods 

### 2.1. Study Design

For the evaluation of the two commercial MAC-ELISA kits we followed two approaches using two distinct panels of patients.

In panel A, we aimed to compare the results of the 2 ELISA kit with dengue status of the patients, dengue positive or dengue negative. Dengue positive patients were selected based on positive DENV probe-based real-time RT-PCR (RT-qPCR). Dengue negative patients were selected based on negative DENV RT-qPCR and positive qPCR for another pathogen. For each kit, sensitivity and specificity was calculated against the RT-qPCR results as the “gold standard”. 

However, this approach leads to selection bias with a set of patients not representative of real situations for dengue diagnosis, so we completed it by a second approach using panel B. 

In panel B, with no reference serological or PCR results, we compared the results of the 2 ELISA kits when used on a representative set of patients admitted to hospital with suspicion of dengue infection. Agreements between the results of the two kits were calculated. 

### 2.2. Patients

The study was conducted at Mahosot Hospital, Vientiane Capital, Laos, between 2008 and 2015. Included patients were patients admitted to hospital with clinical presentation meeting WHO criteria for dengue [14], based on physician judgement. Venous blood was collected on admission and at convalescence and then immediately centrifuged. Serum and buffy coat samples were stored at −80 °C for subsequent investigations. Routine dengue diagnosis consisted of a combination of molecular and serology assays: DENV RT-qPCR as previously described [15], NS1 (Panbio Dengue Early ELISA Cat. E-DEN02P, Alere Inc, Waltham, MA, USA) and anti-dengue IgM ELISAs (Panbio^®^ Dengue IgM Capture ELISA Cat. E-DEN01M/E-DEN01M05). For differential diagnosis, buffy coat samples underwent testing for *Leptospira* [16,17,18], *Orientia tsutsugamushi* and *Rickettsia* spp. qPCRs [19]. 

### 2.3. Ethics Statement

Written informed consent was obtained from all recruited patients or responsible guardians. Ethics approval was obtained from the Lao National Ethics Committee for Health Research and the Oxford Tropical Research Ethics Committee (OXTREC 006-07, 31 January 2008).

### 2.4. Sample Selection

#### 2.4.1. Panel A

65 paired sera (acute serum and its corresponding convalescent serum) were selected based on previous RT-qPCR results, which is a recognized “gold standard” for acute dengue infection diagnosis [7]. Of these, 54 acute samples were positive by DENV RT-qPCR and were classified as dengue infection group. The non-dengue infection group was constituted of 11 patients, 4 positive by qPCR for *Leptospira spp.*, 5 for *O. tsutsugamushi* and 2 for *Rickettsia* spp. All acute samples from non-dengue infection group were all tested negative by DENV RT-qPCR. 

#### 2.4.2. Panel B

Paired sera from 74 patients were randomly selected from 319 consecutive patients. The consecutive patients admitted to hospital for suspicion of dengue infection based on clinical presentation, recruited in 2008 independently of previous laboratory assay results, were assigned numbers from 0 to 318. Every fourth patient was included in the study. 

### 2.5. Anti-Dengue IgM ELISAs

All sera were tested using both Panbio^®^ Dengue IgM Capture ELISA (Alere now Abbott, MA, USA) and DEN Detect^TM^ MAC-ELISA (InBios kit, InBios International, Inc., Seattle, WA, USA) kits following respective manufacturer’s instructions. 

Both kits used 1:100 diluted serum on a microplate coated with anti-human IgM antibodies, then adding dengue antigen and DENV-specific monoclonal antibody labeled with enzyme horseradish peroxidase (Mab/HRP), in a succession of incubations and washings. However, in Panbio process, the complex DENV1-4 antigens – Mab/HRP is assembled in a separate incubation, before being added as a single step into the plate. In the InBios process, serum is loaded in duplicate in two wells, one well is then loaded with Dengue-derived recombinant antigens (DENRA) and the other one with normal cell antigen (NCA) for subtraction of background signal. For both kits, revelation is done by using Tetramethylbenzidine (TMB), stopped after 10 min incubation in the dark. Absorbance is read at 450 nm with a Microplate reader. 

For the Panbio kit, the absorbance average for the calibrator tested in triplicate was calculated then multiplied by the calibration factor to obtain the cut-off value. The Panbio Units (PU) are calculated for each tested serum by multiplying by 10 the ratio: absorbance of tested sample/cut-off value. The result is interpreted as followed: PU < 9 as negative, PU between 9 and 11 as equivocal, and PU > 11 as positive. 

For the InBios kit, immune status ratio (ISR) was determined by calculating the ratio of the DENRA absorbance over the NCA absorbance of the tested serum. An ISR < 1.65 is interpreted as negative, ISR between 1.65 and 2.84 as equivocal, and ISR > 2.84 as positive. 

All equivocal samples were retested and the latter test result was considered as the final result. For analysis purposes, the equivocal results were classified as anti-dengue IgM negative.

### 2.6. Statistical Analysis

For Panel A, sensitivity, specificity, negative predict values (NPV), and positive predictive value (PPV) with 95% confidence intervals (CI) of the ELISA kits for the detection of acute dengue infection were calculated against the “gold standard” RT-qPCRs’ results [20].

In Panel B, without “gold standard” tests, the results obtained with the two ELISA kits were compared by calculating the positive, negative and overall percent agreements as recommended by FDA’s Guidance on Reporting Results from Studies Evaluating Diagnostic Tests [21]. Agreement between the two ELISA test results were also assessed by calculating Cohen’s Kappa coefficient (k), interpreted as follows: values ≤0 as indicating no agreement and 0.01–0.20 as none to slight, 0.21–0.40 as fair, 0.41–0.60 as moderate, 0.61–0.80 as substantial, and 0.81–1.00 as almost perfect agreement [22]. 

All the statistical calculations were performed with RStudio software (Version 1.0.136–© 2009-2016 RStudio, Inc. Boston, MA, USA) [23].

## 3. Results

### 3.1. Panel A

65 febrile patients were selected according to their qPCR results and classified in two groups, 54 patients in the dengue group and 11 patients in the non-dengue group. The median days (IQR) of fever onset for the dengue group was 5 (4–6), and 7.5 (5.5–9.75) for the non-dengue group (Table 1). The median days (IQR) between admission and convalescent samples were 12 (9–14) and 3 (2.25–6.25) for the dengue and non-dengue groups, respectively. 

Of the 54 patients from dengue group, anti-dengue IgM was positive in 35 (64.8%) admission samples and in 39 (72.2%) convalescent samples by the Panbio kit, and in 10 (18.5%) admission sera and in 39 (72.2%) convalescent sera by the InBios kit (Table 2 and Table 3). When the results from admission and convalescent sera were combined, the Panbio kit was positive for anti-dengue IgM in 47 (87%) patients and the InBios kit in 39 (72.2%) patients. 

For the 11 patients from the non-dengue group, Panbio and InBios kits were both negative for all admission sera. All convalescent sera were tested negative by InBios kit but three were found positive by Panbio kit. 

When only based on the results from admission sera, the sensitivity and specificity of the Panbio kit were 64.8% (95%CI: 50.6–77.3%) and 100% (95%CI: 71.5–100%), respectively. The sensitivity and specificity of the InBios kit were 18.5% (95%CI: 9.3–31.4%) and 100% (95%CI: 71.5–100%), respectively.

When the results of admission and convalescent samples were combined, the sensitivity of the kits increased, to 87% (95%CI: 75.1–94.6%) for Panbio kit and to 72.2% (95%CI: 58.4–83.5%) for InBios kit, whereas the specificity decreased for Panbio kit to 72.7% (95%CI: 39–93.9%), and the specificity for InBios kit remained at 100% (95%CI: 71.5–100%). 

### 3.2. Panel B

74 patients with paired sera were randomly selected from 319 patients admitted to Mahosot Hospital with suspicion of dengue infection throughout 2008 (Figure 1). The median (IQR) days of fever onset on admission was 5 (3–6), and the median (IQR) days between admission and convalescent sample collection was 8 (6–9.8). 

Of 74 admission samples, 16 (21.6%) were found positive by the Panbio ELISA, and 9 (12.2%) by the InBios ELISA. Of 74 convalescent samples, 15 (20.3%) were found positive by the Panbio kit, and 8 (10.8%) by the InBios kit. When combining the results from admission and convalescent samples, 21 (28.4%) patients were found positive by the Panbio kit, and 12 (16.2%) by the InBios kit. 

The calculations of percent agreements and Cohen’s kappa coefficient between the two ELISA kits are summarized in Table 4. The positive percent agreement (95% CI) between the results of the two ELISA kits from admission samples was low, 31.2% (11.0–58.7%) and slightly increased when comparing results from convalescent sera, 53.3% (26.6–78.7%), or combination of results from admission and convalescent sera, 47.6% (25.7–70.2%). High negative percent agreement (95% CI) was observed between the two ELISA kits, of 93.1% (83.3–98.1%) on admission sera, 100% (93.9–100%) on convalescent sera and 96.2% (87.0–99.5%) when combining both admission and convalescent results. Cohen’s kappa coefficient (95%CI) between the two ELISA kits was 0.29 (0.025–0.55) for admissions sera, 0.65 (0.41–0.88) for convalescent sera and 0.5 (0.28–0.73) when combining results from admission and convalescent sera.

## 4. Discussion

In this study, we observed fair agreement (κ = 0.29) between the results obtained with the Panbio^®^ Dengue IgM Capture ELISA (Alere, now Abbott, Massachusetts, USA) and DEN Detect^TM^ MAC-ELISA (InBios kit, InBios International, Inc., Seattle, WA) kits for the detection of anti-dengue IgM in a set of 74 patients admitted at Mahosot hospital with clinical dengue presentation. The agreement was slightly better (moderate, (κ = 0.5)) when combining results from admission and convalescent sera. When the IgM ELISAs were evaluated on a set of 65 qPCR confirmed patients (54 dengue and 11 other aetiologies), the Panbio kit demonstrated significantly higher sensitivity (64.8%; 95%CI: 50.6–77.3%, *p* = 0.028 using chi-square test) when compared to the InBios kit (18.5%; 95%CI: 9.3–31.4%) when calculated for single acute serum. The sensitivity of the InBios kit increased notably when combining admission and convalescent results, but remained lower than for Panbio kit (87.0% (95%CI: 75.1–94.6%) for Panbio vs. 72.2% (95%CI: 58.4–83.5%) for InBios). However, InBios appeared to have better specificity with none of the 11 “non-dengue” patients found positive, whereas three were found positive using Panbio on convalescent samples. However, the small sample size did not allow to ascertain the statistical significance of the difference between the specificity results.

Accuracy of diagnostic tests are assessed by their sensitivity and specificity. For the same diagnostic test, sensitivity and specificity are varying according to the epidemiological context, hence the need for their evaluation in the same context they are used for diagnosis. This is achieved by evaluating their performance against a reference standard. As defined by STARD (Standards for Reporting of Diagnostic Accuracy), the reference standard or “gold standard” is considered to be the best available method for establishing the presence or absence of the target condition [21]. For dengue diagnosis, laboratory tests to be used are dependent on the stage of the course of the disease; in the early stage of the disease, direct tests can be used, whereas immunological tests are the methods of choice at the end of acute phase [7]. Thus, a combination of direct and indirect tests is necessary to cover all stages of patient presentation. In the Lao setting, no gold standard was available for immunological assay, therefore, we combined two approaches to evaluate the performance of the ELISA kits for anti-dengue IgM detection in the context of diagnosis in Lao setting. We established the patients’ infection status “dengue” and “non-dengue” based on the PCR results which is a recognized “gold standard” [7], with the “non-dengue patients” being positive for another aetiology. However, this biased the selection of patients to the early stage of the disease. Therefore, this set of patients, panel A, was not representative of the dengue patients seeking care in a later stage of infection. For that reason, we established the panel B consisting of patients admitted to Mahosot Hospital with dengue presentation, as a representation of real situation of dengue diagnosis in Laos. In the absence of gold standard, we compared the performance of the two commercial MAC-ELISA kits by measuring agreements.

The sensitivity (87%, (75.1–94.6)) and specificity (72.2%, (39–93.9)) for the Panbio kit on combined sera evaluated in this study were in concordance compared to other studies (Groen et al. [9] (sensitivity: 87%, specificity: 96%); Blacksell et al. [8] (sensitivity: 88.6%, specificity: 87.8%); Pal et al. [11] (sensitivity: 87.6%, specificity: 88.1%)). The sensitivity (72.2%, (58.4–83.5)) for the InBios kit on combined sera was lower than the other two previous evaluation studies (Namekar et al. [13] (sensitivity: 92%, specificity: 94%); Welch et al. [12] (sensitivity: 88.7%, specificity: 93.1%)), with similar specificity results. The lower sensitivity demonstrated in this study might be due to the small sample size, the uncertain status of dengue infection (primary or secondary), and the choice of “gold standard”.

Thus, the results from panel A suggest that the Panbio kit would be more sensitive but less specific than the InBios kit for the detection of anti-dengue IgM. This is supported by the observation of increased numbers of positive patients between admission and convalescent samples when tested using InBios kit, suggesting that InBios kit would have better performance when the rate of IgM is higher. This is also supported by the fact that when comparing patients with sample collected before 5 days of fever to patient with sample collected after 5 days of fever, highest sensitivity was observed in the latest group (data not shown). The testing results from panel B were in concordance with the results from panel A with percent agreements, when comparing InBios results to Panbio results, of 31.3% (95%CI: 11.0–58.7%) for admission samples and 53.3% (95%CI: 26.6–78.7%) for convalescent samples. 

During an acute dengue infection, the detectable levels of anti-dengue IgM antibodies start at ~5 days after fever onset and keep increasing until 3 days after defervescence, and the IgM antibodies persist in the blood for ~90 days [24]. Our results showed that the sensitivities and agreement of Panbio and InBios kits are better on paired sera than on acute samples only. According to the antibody response to dengue infection, during an early stage dengue infection, there might not have enough IgM antibodies to be detected, which could lead to the false negative results by the anti-dengue IgM ELISA kits. In real life, the physicians often make the diagnosis based on single sample’s results, therefore, the diagnosis of dengue infection based on anti-dengue IgM ELISA results should be combined with direct tests to confirm the infections. Moreover, in the context of use of acute sera, Panbio kit should probably be preferred over the less sensitive InBios kit. 

Limitations of the study include that the sample size is relatively small. For both panels, we do not know whether the patients had primary or secondary dengue infection, or if DENV serotype would play a role in ELISA kits performance or the possibility of infection with other flavivirus could not be excluded. 

Despite all of the above limitations, our results suggest that the Panbio^®^ Dengue IgM Capture ELISA has increased sensitivity, especially when only acute sera are available. There are multiple ELISAs used for dengue diagnosis but very few comparisons of the diagnostic accuracy between ELISAs [25]; more such investigations are needed to understand how these differences affect our understanding of dengue epidemiology. 

## Figures and Tables

**Figure 1 tropicalmed-04-00111-f001:**
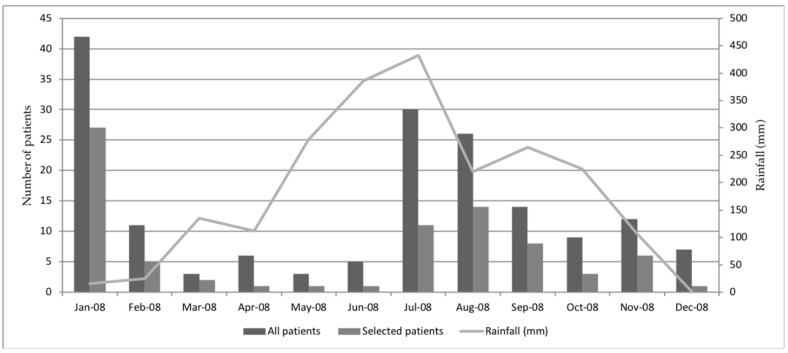
Distribution over time of patient sample received in the laboratory in 2008 for suspicion of dengue infection. Patient with suspicion of dengue infection are patients with clinical signs meeting dengue criteria following WHO guideline [14].

**Table 1 tropicalmed-04-00111-t001:** Summary of dengue suspected patients’ samples in Panel A (infection status by PCRs’ results, number, fever days, interval between admission and convalescence).

Patient Status (based on Corresponding PCR Results)		n	Days of Fever on Admission, Median (IQR)	Days between Admission and Convalescence, Median (IQR)
Dengue infection	Dengue	54	5 (4–6)	12 (9–14)
Non-dengue infection	*Leptospira* spp.	4	5 (2.75–7.5)	2.5 (1.75–4.25)
	*O. tsutsugamushi*	5	7 (5–8)	3 (2.75–3.25)
	*Rickettsia* spp.	2	7 (7–10)	8 (7.5–8.5)
	Total	65	5 (4–6)	11 (6–14)

Dengue infection: serum sample with a positive result for dengue real-time RT-PCR. Non-dengue infection: confirmed dengue RT-PCR negative serum sample with a positive molecular result for another pathogen (as listed in the table).

**Table 2 tropicalmed-04-00111-t002:** Accuracy of Panbio ELISA for dengue diagnosis for patients from Panel A.

		Admission Serum				Convalescent Serum				Combined Sera Result			
		Dengue Status		Total	Diagnostic Accuracy, % (95% CI)	Dengue Status		Total	Diagnostic Accuracy, % (95% CI)	Dengue Status		Total	Diagnostic Accuracy, % (95% CI)
		Positive	Negative			Positive	Negative			Positive	Negative		
					Sen: 64.8 (50.6–77.3)				Sen: 77.8 (64.4–87.9)				Sen: 87.0 (75.1–94.6)
Panbio kit	Positive	35	0	35	Spe: 100 (71.5–100)	39	3	42	Spe: 72.7 (39–93.9)	47	3	50	Spe: 72.7 (39–93.9)
	Negative	19	11	30	PPV: 100 (89.9–100)	15	8	23	PPV: 93.3 (81.7–98.6)	7	8	15	PPV: 94 (83.5–98.7)
	Total	54	11	65	NPV: 36.7 (19.9–56.1)	54	11	65	NPV: 40.0 (19.1–63.9)	54	11	65	NPV: 53.3 (26.6–78.7)

Panbio kit: Panbio^®^ Dengue IgM Capture ELISA (Panbio Kit, Alere, Waltham, MA, USA). The sensitivity, specificity, PPV, and NPV were calculated based on dengue real-time RT-PCR gold standard, all non-dengue infected samples were confirmed by other positive PCR results.

**Table 3 tropicalmed-04-00111-t003:** Accuracy of InBios ELISA for dengue diagnosis for patients from Panel A.

		Admission Serum				Convalescent Serum				Combined Sera Result			
		Dengue Status		Total	Diagnostic Accuracy, % (95% CI)	Dengue Status		Total	Diagnostic Accuracy, % (95% CI)	Dengue Status		Total	Diagnostic Accuracy, % (95% CI)
		Positive	Negative			Positive	Negative			Positive	Negative		
					Sen: 18.5 (9.2–31.4)				Sen: 72.2 (58.4–83.5)				Sen: 72.2 (58.4–83.5)
InBios kit	Positive	10	0	10	Spe: 100 (71.5–100)	39	0	39	Spe: 100 (71.5–100)	39	0	39	Spe: 100 (71.5–100)
	Negative	44 *	11	55	PPV: 100 (69.2–100)	15 *	11	26	PPV: 100 (91–100)	15	11	26	PPV: 100 (91–100)
	Total	54	11	65	NPV: 20 (10.4–33)	54	11	65	NPV: 42.3 (23.4–63.1)	54	11	65	NPV: 42.3 (23.4–63.1)

InBios kit: DEN Detect^TM^ MAC-ELISA (InBios kit, InBios International, Inc., Seattle, WA, USA). 11 admission serum, 5 convalescent serum analyzed by InBios kit persisted equivocal results after a second test. The sensitivity, specificity, PPV, and NPV were calculated based on dengue real-time RT-PCR gold standard, all non-dengue infected samples were confirmed by other positive PCR results. * 2 admission serum analyzed by InBios kit and 1 convalescent serum analyzed by Panbio kit persisted equivocal results after a second test.

**Table 4 tropicalmed-04-00111-t004:** Comparison of the Panbio and InBios kits results for dengue suspected patients, Panel B.

		Admission Serum			
		Panbio kit			Percent Agreement, % (95% CI)
		Positive	Negative	Total	
**InBios kit**	Positive	5	4	9	Positive: 31.3 (11.0–58.7)
	Negative	11	54	65	Negative: 93.1 (83.3–98.1)
	Total	16	58	74	overall: 79.7 (68.9–88.2)
		Cohen’s kappa index (k) = 0.29 (0.025–0.55)			
		**Convalescent serum**			
		**Panbio kit**			**Percent Agreement, % (95% CI)**
		**Positive**	**Negative**	**Total**	
**InBios kit**	Positive	8	0	8	Positive: 53.3 (26.6–78.7)
	Negative	7	59	66	Negative: 100 (93.9–100)
	Total	15	59	74	overall: 90.5 (81.5–96.1)
		Cohen’s kappa index (k) = 0.65 (0.41–0.88)			
		**Combined admission and convalescent results**			
		**Panbio kit**			**Percent Agreement, % (95% CI)**
		**Positive**	**Negative**	**Total**	
**InBios kit**	Positive	10	2	12	Positive: 47.6 (25.7–70.2)
	Negative	11	51	62	Negative: 96.2 (87.0–99.5)
	Total	21	53	74	overall: 82.4 (71.8–90.3)
		Cohen’s kappa index (k) = 0.5 (0.28–0.73)			

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
