# Peer review of "Comparison of Two Commercial ELISA Kits for the Detection of Anti-Dengue IgM for Routine Dengue Diagnosis in Laos"

_tropicalmed, 2019, doi:10.3390/tropicalmed4030111_

Round 1

Reviewer 1 Report

This is a useful comparison of two commercial dengue IgM tests. The structure of the manuscript is good and the writing is generally clear although I have a few suggestions for correcting some English expression.

The authors correctly point out that the sensitivity of the IgM tests will vary according to the interval between illness onset and when blood was taken. I would be curious as to whether serum collected before five days and after five days showed a difference in sensitivity in panel A (maybe include in the discussion). I do not think the negative agreement rate in panel B (said to be 100%) can be correct when there were seven negative tests on the in BIOS kit that were positive on the pan bio kit - could you recalculate that?

For the minor corrections I will use the line numbers that appeared on the PDF
line 42 - suggest "into" rather than "in"

line 43-omit "of" between 2.5% and mortality

line 45 omit the "..." In the parentheses

line 64 replace "was proceeded in" with "proceeded with" and "based on" with "on the basis of"

line 86 replace "in" with "of"

line 89/ 90 replace "were submitted to" with "underwent testing for"...... using qPCR

line 109 last sentence, suggest reword "every fourth patient was included in the study"

line 147 and 149 "days (IQR)" 

there is inconsistent formatting in the spaces for reference 1-8 compared to the rest of the references

Reviewer 2 Report

Comments to authors: This manuscript presents an interesting direct comparison of two commercial ELISA IgM kits for detection of dengue-specific antibodies. The use of two different patient populations was to be commanded. The title seems to suggest that the goal was to compare the two kits to find one that is better for routine dengue diagnosis in the context of Laos. Therefore, the first population could be considered as a validation of the performance of the two kits and the results suggests that PanBio’s kit is probably a better choice. The 2nd population seems to intend to really test how the two assays perform in “real situation” when a patient comes in with suspicion of dengue based on WHO criteria. So the idea seems to be testing whether one kit performs better to “diagnose” dengue in this particular population. This is particularly true if any “final diagnosis made” on these patients can be used as a reference to evaluate the two assays independently. It is not clear to this reviewer why this analysis was not attempted. Specific comments are: Line 90: Are these tests only apply to patients in Panel A? It seems to be a more general statement that is applicable to both Panels. If patients in both panels have gone through this "routine dengue diagnosis", can the panel B data be analyzed using the final diagnosis as the "reference test"? In other words, the combined results would give a "diagnosis" of the suspected dengue patients included in panel B a final diagnosis. Since the samples are randomly selected, it is possible to ask whether PanBio or InBio's IgM ELISA is better to diagnose dengue infection based on a single acute sample. If the data is available, can this be done? Line 130: Does this mean that all equivocal samples were tested twice only and the 2nd outcome is the reported outcome? What if the 2nd outcome is contradictory to the first? What's the rationale to take the "latter" result as the final result? Line 154: Maybe it is a good idea to state that these "non-dengue infections" are confirmed dengue RT-PCR negative here although it was stated earlier in the methods. Line 267: rephrase this sentence. Redundant expression.

Round 2

Reviewer 1 Report

There were some slight discrepencies in response letter vs version attached - editorial staff will fix- otherwise I am satisfied with responses

Reviewer 2 Report

I appreciated authors effort to address those questions raised. I accepted the reasoning and modifications made to the manuscript.